# Integrating Episodic and Global Novelty Bonuses for Efficient Exploration

## Abstract

Exploration in environments which differ across episodes has received increasing attention in recent years. Current methods use some combination of *global novelty bonuses*, computed using the agent's entire training experience, and *episodic novelty bonuses*, computed using only experience from the current episode. However, the use of these two types of bonuses has been ad-hoc and poorly understood. In this work, we first shed light on the behavior these two kinds of bonuses on hard exploration tasks through easily interpretable examples. We find that the two types of bonuses succeed in different settings, with episodic bonuses being most effective when there is little shared structure between environments and global bonuses being effective when more structure is shared. We also find that combining the two bonuses leads to more robust behavior across both of these settings. Motivated by these findings, we then investigate different algorithmic choices for defining and combining function approximation-based global and episodic bonuses. This results in a new algorithm which sets a new state of the art across 18 tasks from the MiniHack suite used in prior work. Our code is public at `web-link`.

## 1 Introduction

Balancing exploration and exploitation is a long-standing challenge in reinforcement learning (RL). A large body of research has studied this problem within the Markov Decision Processes (MDP) framework (Sutton & Barto, 2018), both from a theoretical standpoint (Kearns & Singh, 2002; Brafman & Tennenholtz, 2002; Agarwal et al., 2020) and an empirical one. This has led to practical exploration algorithms such as pseudocounts (Bellemare et al., 2016b), intrinsic curiosity modules (Pathak et al., 2017) and random network distillation (Burda et al., 2019), yielding impressive results on hard exploration problems like Montezuma's Revenge and PitFall (Bellemare et al., 2012).

More recently, there has been increasing interest in algorithms which move beyond the MDP framework. The standard MDP framework assumes that the agent is initialized in the same environment at each episode (we will refer to these MDPs as *singleton* MDPs). However, several studies have found that agents trained in singleton MDPs exhibit poor generalization, and that even minor changes to the environment can cause substantial degradation in agent performance (Zhang et al., 2018b; Justesen et al., 2018; Zhang et al., 2018a; Cobbe et al., 2019; Kirk et al., 2021a). This has motivated the use of *contextual* MDPs (CMDPs, (Hallak et al., 2015)), where different episodes correspond to different environments which nevertheless share some structure. Examples of CMDPs include procedurally-generated environments (Chevalier-Boisvert et al., 2018; Samvelyan et al., 2021; Küttler et al., 2020; Juliani et al., 2019; Cobbe et al., 2020; Beattie et al., 2016; Hafner, 2021; Petrenko et al., 2021) or embodied AI tasks where the agent must generalize across different physical spaces (Savva et al., 2019; Shen et al., 2020; Gan et al., 2020; Xiang et al., 2020).

While exploration is well-studied in the singleton MDP case, it becomes more nuanced when dealing with CMDPs. For singleton MDPs, a common and successful strategy consists of defining an exploration bonus which is added to the reward function being optimized. This exploration bonus typically represents how novel the current state is, where novelty is computed with respect to the entirety of the agent's experience across all episodes. However, it is unclear to what extent this strategy is applicable in the CMDP setting—if two environments corresponding to different episodes are very different, we might not want the experience gathered in one to affect the novelty of a state observed in the other.

An alternative to using global bonuses is to use episodic ones. Episodic bonuses define novelty with respect to the experience gathered in the current episode alone, rather than across all episodes. Recently, several works (Stanton & Clune, 2018; Raileanu & Rocktäschel, 2020; Flet-Berliac et al., 2021; Zhang et al., 2021b; Henaff et al., 2022) have used episodic bonuses, with Henaff et al. (2022) showing that this is an essential ingredient for solving sparse reward CMDPs. However, as we will show, an episodic bonus alone may not be optimal if there is considerable shared structure across different episodes in the CMDP.

In this work, we study how to best define and integrate episodic and global novelty bonuses for exploration in CMDPs. First, through a series of easily interpretable examples using episodic and global count-based bonuses, we shed light on the strengths and weaknesses of both types of bonuses. In particular, we show that *global bonuses, which are commonly used in singleton MDPs, can be poorly suited for CMDPs that share little structure across episodes; however, episodic bonuses, which are commonly used in contextual MDPs, can also fail in certain classes of singleton MDPs where knowledge transfer across episodes is crucial.* Second, we show that by multiplicatively combining episodic and global bonuses, we are able to get robust performance on both contextual MDPs that share little structure across episodes and singleton MDPs that are identical across episodes. Third, motivated by these observations, we comprehensively evaluate different combinations of episodic and global bonuses which do not rely on counts, as well as strategies for integrating them, on a wide array of tasks from the MiniHack suite (Samvelyan et al., 2021). Our investigations yield a new algorithm which combines the elliptical episodic bonus of Henaff et al. (2022) and the NovelD global bonus of Zhang et al. (2021b), which sets a new state of the art across 18 tasks from the MiniHack environment, solving the majority of them. Our code is available at `web-link`.

## 2 BACKGROUND

### 2.1 CONTEXTUAL MDPS

We consider a contextual Markov Decision Process (CMDP) defined by $(\mathcal{S}, \mathcal{A}, \mathcal{C}, P, r, \mu_C, \mu_S)$ where $\mathcal{S}$ is the state space, $\mathcal{A}$ is the action space, $\mathcal{C}$ is the context space, $P$ is the transition function, $\mu_S$ is the initial state distribution conditioned on context and $\mu_C$ is the context distribution. At each episode, we first sample a context $c \sim \mu_C$ and an initial state $s_0 \sim \mu_S(\cdot|c)$. At each step $t$ in the episode, the next state is then sampled according to $s_{t+1} \sim P(\cdot|s_t, a_t, c)$ and the reward is given by $r_t = r(s_t, a_t, c)$. Let $d_\pi^c$ represent the distribution over states induced by following policy $\pi$ with context $c$. The goal is to learn a policy which maximizes the expected return, averaged across contexts:

$$R = \mathbb{E}_{c \sim \mu_C, s \sim d_\pi^c, a \sim \pi(\cdot|s)}[r(s, a)]$$

Examples of CMDPs include procedurally-generated environments, such as ProcGen (Cobbe et al., 2020), MiniGrid (Chevalier-Boisvert et al., 2018), NetHack (Küttler et al., 2020), or Mini-Hack (Samvelyan et al., 2021), where each context $c$ corresponds to the random seed used to generate the environment; in this case, the number of contexts $|\mathcal{C}|$ is effectively infinite (we will slightly abuse notation and denote this case by $|\mathcal{C}| = \infty$). Other examples include embodied AI environments (Savva et al., 2019; Szot et al., 2021; Gan et al., 2020; Shen et al., 2020; Xiang et al., 2020), where the agent is placed in different simulated houses and must navigate to a location or find an object. In this setting, each context $c \in \mathcal{C}$ represents a house identifier and the number of houses $|\mathcal{C}|$ is typically between 20 and 1000. More recently, CARL (Benjamins et al., 2021) was introduced as a benchmark for testing generalization in contextual MDPs. However, their focus is on using privileged information about the context $c$ to improve generalization, which we do not assume access to here. For an in-depth review of the literature on CMDPs and generalization in RL, see Kirk et al. (2021b). Singleton MDPs are a special case of contextual MDPs with $|\mathcal{C}| = 1$.

### 2.2 EXPLORATION BONUSES

At a high level, exploration bonuses operate by estimating the novelty of a given state, and assigning a high bonus if the state is novel according to some measure. The exploration bonus is then combined with the extrinsic reward provided by the environment, and the result is optimized using RL. More precisely, the reward function optimized by the agent is given by:

$$\bar{r}(s, a) = r(s, a) + \alpha \cdot b(s, a) \qquad (1)$$

where $r(s, a)$ is the extrinsic reward, $b(s, a)$ is the exploration bonus, and $\alpha$ is a parameter governing the balance between exploration and exploitation. Some bonuses do not depend on $a$ or additionally depend the next state $s'$, which will be clear from the context. To account for the sometimes wide variation in the scale of exploration bonuses across different environments and different points in time during training, the exploration bonus is sometimes divided by a running estimate of its standard deviation, as is done in Burda et al. (2019).

In tabular domains with a small number of discrete states, a common choice is to use the inverse counts: $b(s) = 1/\sqrt{N(s)}$ (Strehl & Littman, 2006), where $N(s)$ is the number of times state $s$ has been encountered by the agent. However, in many settings of interest the number of possible states is large or infinite, and many states will not be seen more than once, rendering this bonus ineffective since all states will be rewarded by the same amount. This has motivated alternative approaches using function approximation. The methods below have proven successful on sparse reward singleton MDPs (RND) and/or sparse reward CMDPs (RIDE, AGAC, NovelD and E3B).

**Random Network Distillation (RND)** (Burda et al., 2019) randomly initializes a neural network $\bar{f} : \mathcal{S} \to \mathbb{R}^k$, and then trains a second neural network $f$ with the same architecture to predict the outputs of $\bar{f}$ on states encountered by the agent. The exploration bonus associated with a given state $s$ is given by the MSE:

$$b_{\mathrm{RND}}(s_t) = \|f(s_t) - \bar{f}(s_t)\|_2^2 \qquad (2)$$

The intuition is that for states similar to ones previously encountered by the agent, the error will be low, whereas it will be high for very different states. RND has performed well on hard singleton MDPs and is a commonly used component of other exploration algorithms.

**Novelty Difference (NovelD)** (Zhang et al., 2021b) uses the difference between RND bonuses at two consecutive time steps, regulated by an episodic count-based bonus. Specifically, its bonus is:

$$b_{\mathrm{NovelD}}(s_t, a, s_{t+1}) = \Big[ b_{\mathrm{RND}}(s_{t+1}) - c \cdot b_{\mathrm{RND}}(s_t) \Big]_+ \cdot \mathbb{I}[N_e(s_{t+1}) = 1] \qquad (3)$$

Here $b_{\mathrm{RND}}$ represents the RND bonus defined above, and $N_e(s)$ represents the number of times $s$ has been encountered within the current episode. The first term is a *global novelty bonus*, which measures novelty with respect to cross-episode experience, whereas the second term is an *episodic novelty bonus*, which measures novelty with respect to experience within the current episode only.

**Adversarially Guided Actor-Critic (AGAC)** (Flet-Berliac et al., 2021) also combines global and episodic novelty bonuses. Its bonus is defined by:

$$b_{\mathrm{AGAC}}(s_t) = D_{\mathrm{KL}}(\pi(\cdot|s_t)\|\pi_{\mathrm{adv}}(\cdot|s_t)) + \beta \frac{1}{\sqrt{N_e(s_t)}} \qquad (4)$$

where $\pi_{\mathrm{adv}}$ is a policy trained to mimic the behavior policy $\pi$ (usually with a smaller learning rate). The motivation is that this will encourage the policy to adopt different behaviors as it tries to remain different from the adversary. The second term is an episodic bonus based on $N_e(s)$, the number of times the state $s$ has been encountered within the current episode.

**Rewarding Impact-Driven Exploration (RIDE)** (Raileanu & Rocktäschel, 2020) uses an episodic novelty bonus which is the product of two terms: a count-based reward and the difference between two consecutive state embeddings:

$$b_{\mathrm{RIDE}}(s_t) = \frac{1}{\sqrt{N_e(s_t)}} \|\phi(s_{t+1}) - \phi(s_t)\|_2 \qquad (5)$$

Here the $\phi$ embedding is learned using a combination of inverse and forward dynamics models. The motivation for the second term in the bonus is to reward the agent for taking actions which cause significant changes in the environment. RIDE does not use a global novelty bonus.

**Exploration via Elliptical Episodic Bonuses (E3B)** (Henaff et al., 2022) also uses an episodic novelty bonus only, and is motivated by the following observation: while the count-based episodic bonuses used in NovelD, RIDE and AGAC are essential for good performance, they do not scale to

complex environments where a given state is unlikely to be seen more than once within an episode. E3B uses a feature extractor $\phi$ learned using an inverse dynamics model, and defines the episodic bonus as follows:

$$b_{\text{E3B}}(s_t) = \phi(s_t)^\top C_{t-1}^{-1} \phi(s_t), \qquad C_{t-1} = \sum_{i=t_0}^{t-1} \phi(s_i)\phi(s_i)^\top + \lambda I \qquad (6)$$

Here $t_0$ denotes the start of the current episode. This can be seen as a generalization of an episodic count-based bonus to continuous state spaces.

## 3    WHEN ARE EPISODIC AND GLOBAL NOVELTY BONUSES USEFUL?

Although RIDE, NovelD, AGAC and E3B all use different combinations of episodic and global novelty bonuses, their use in CMDPs has been largely heuristic. The RIDE and NovelD papers simply state that the episodic bonus is included to prevent the agent from going back and forth between a sequence of states within the same episode. Furthermore, the global novelty bonuses are justified using the singleton MDP case, but it is unclear to what extent these justifications carry over to the CMDP case. Therefore, a closer investigation of when episodic and global novelty bonuses are useful in CMDPs is required. Details for all experiments in this section are in Appendix D.

### 3.1    ADVANTAGES OF EPISODIC BONUSES

We begin by providing an example of CMDPs where global novelty bonuses fail and episodic bonuses succeed. Consider the procedurally-generated MiniHack environment shown in Figure 1. Here, each episode corresponds to a different map where the agent must navigate from the starting location to the goal. The agent only receives reward if it reaches the goal, and the episode terminates if it touches the walls which are made of lava. Because of this, random exploration has a very small chance of reaching the goal before the episode ends, and exploration bonuses are needed.

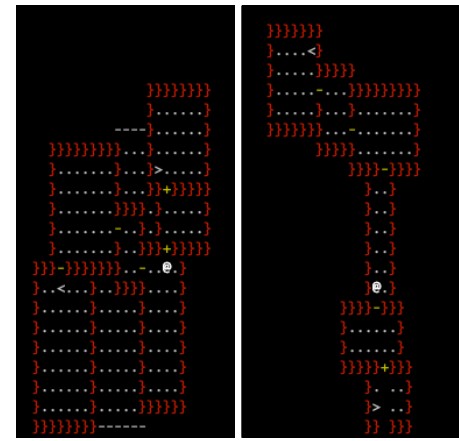

Figure 1:   Two different contexts of the `MultiRoom-N6-Lava` environment. Legend: `@`: agent, `<`: start, `>`: goal, `}`: lava

We ask the question: are global or episodic novelty bonuses more appropriate here? For simplicity, we consider bonuses based on counts of $(x, y)$ locations, which are commonly used in prior work (Flet-Berliac et al., 2021; Samvelyan et al., 2021; Zhang et al., 2021b) to avoid the issue of each state being unique:

$$b_{\text{global}}(s) = \frac{1}{\sqrt{N(\psi(s))}}, \qquad b_{\text{episodic}}(s) = \mathbb{I}[N_e(\psi(s)) = 1]^1 \qquad (7)$$

Here $N$ represents counts over all the agent's experience, and $N_e$ represents counts only within the current episode, while $\psi$ is a feature extractor which extracts the $(x, y)$ coordinates of the agent from the state. In general, methods which do not require handcrafted features are preferable, and we focus on them in Section 4. However, this simple bonus facilitates interpretability, which is the focus of this section.

Using the global novelty bonus encourages the agent to learn a sequence of policies which, together, cover all the $(x, y)$ locations. This is appropriate when the map is the same across all episodes: eventually, one of the policies will cover the $(x, y)$ location corresponding to the goal and the agent will receive reward. However, this is not appropriate in this CMDP setting where the environment

---

[1]We also tried $\frac{1}{\sqrt{N_e(s)}}$ for the episodic bonus, but it didn't work as well.

changes each episode. To see this, note since the goal location changes along with the rest of the map each episode, it can find itself in a region whose global bonus has already been depleted. The more uniform the goal sampling is, the less likely it is the global bonus will be useful.

In contrast, the episodic bonus encourages the agent to visit as many $(x, y)$ locations as possible within each episode. Since each episode corresponds to its own map, a well-optimized policy will thus have a high chance of visiting a region close to the goal, which in turn will increase the chance of reaching the goal and receiving reward.

To verify this argument, we train agents using the global and episodic bonuses in equation 7 over different numbers of contexts $|\mathcal{C}|$ on the `MiniHack-MultiRoom-N6-Lava` environment shown in Figure 1. The number of contexts represents the number of distinct maps, and one of them is chosen at random at the start of each episode. Results are shown in Figure 2. The agent using the global bonus obtains perfect performance for the singleton MDP setting where $|\mathcal{C}| = 1$, but performance steadily degrades as the number of contexts increases. In contrast, when using the episodic bonus, performance remains high as the number of contexts increases. We found that performance remains high ($0.86 \pm 0.1$) even when $|\mathcal{C}| = \infty$ (no two maps are repeated during training).

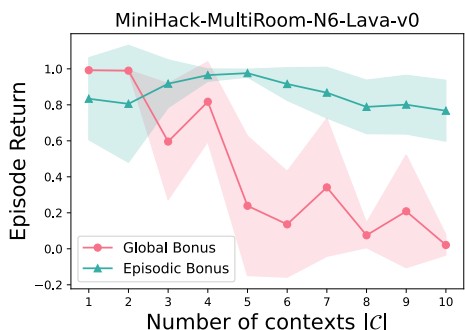

Figure 2: Mean performance for global and episodic count-based exploration bonuses with different numbers of contexts (i.e. maps). Shaded region indicates standard deviation over 5 seeds.

We provide further evidence for this argument with experiments on Habitat, a photorealistic simulator of indoor environments. Habitat is conceptually similar to the `MiniHack-MultiRoom` environment in the sense that at each episode, the agent finds itself in a different indoor space consisting of connected rooms. However, the maps in Habitat are considerably more complex and the observations are pixel-based. Here we compare global and episodic bonuses based on function approximation, since counts are not meaningful with high-dimensional im-

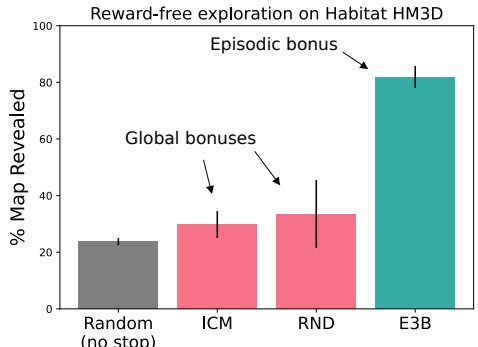

Figure 3: Habitat exploration with $|\mathcal{C}| = 1000$.

ages. We use the reward-free exploration setup described in (Henaff et al., 2022). Results are shown in Figure 3. We see that, similarly to `MiniHack-MultiRoom`, the global bonuses (ICM and RND) perform poorly whereas the episodic bonus (E3B) performs well. See Appendix D for details.

### 3.2 ADVANTAGES OF GLOBAL BONUSES

Does this mean that we should always prefer an episodic bonus to a global one? Unfortunately, the answer is not so clear-cut. We now provide an example where the episodic bonus fails but the global bonus succeeds. Consider a singleton MDP with $M$ corridors which can be crossed in $T$ steps, with a single one containing reward at the end (shown in Figure 4). If the episode length is $T$, then any policy which goes to the end of any of the $M$ corridors will get equivalent episodic bonus, and hence the chance of success will be $1/M$. On the other hand, a global bonus will solve the task: after visiting one of the corridors, the global bonus there will eventually become depleted and the agent will move on to another one, eventually visiting the corridor with the reward.

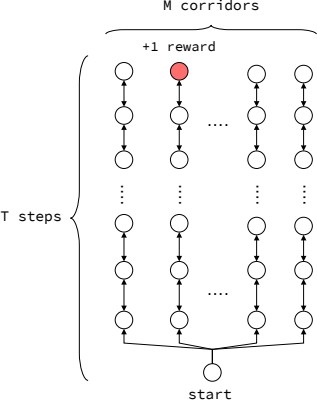

Figure 4: Simple example where episodic bonus fails.

We illustrate this argument using a singleton version of the `MiniHack-Corridors-R5` environment (shown in Figure 5a), where the agent must explore different corridors to find its way to the exit. This is similar to the example in Figure 4 in the sense that the agent will likely need to explore multiple dead ends before finding the goal. Figure 5b shows results for agents trained with the episodic and global bonus. In contrast to the previous example, but consistent with our argument above, the global bonus succeeds across all seeds whereas the episodic bonus produces inconsistent performance across seeds, leading to poor performance overall.

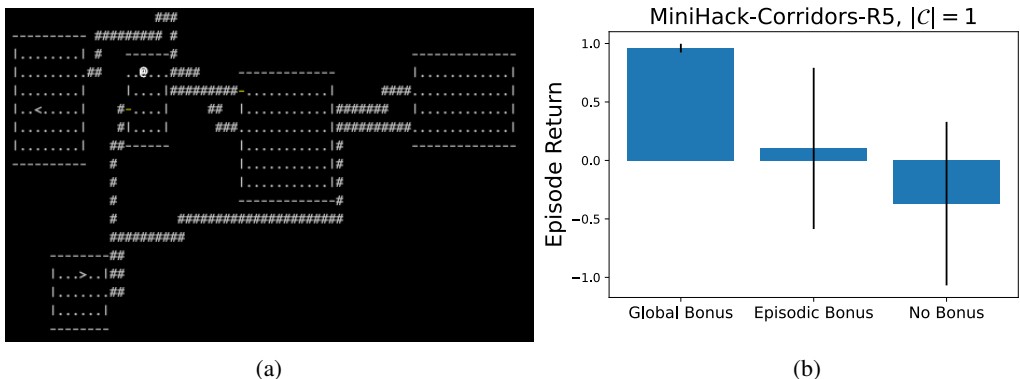

(a)                                                          (b)

Figure 5: a) Example map for `MiniHack-Corridors-R5` enviroment. `@` indicates agent, `#` corridors connecting rooms, `<` start location and `>` goal. b) Performance of agents trained with the global bonus and the episodic bonus on a singleton version of the environment. Error bars represent the standard deviation across 5 random seeds.

Are global bonuses only useful in the special case of singleton MDPs? We next show that this is not the case, and that global bonuses can also be useful in general CMDPs with large $|\mathcal{C}|$, provided they exhibit certain shared structures across contexts. We consider the `MiniHack-KeyRoom-S10` environment, illustrated in Figure 6a. In this environment, the agent must pick up a key and use it to open a door to a small room and reach the exit. Here different contexts correspond to different placements of the agent, key, room, door and exit. A property of this environment is that when the agent visits certain states along the optimal trajectory, it receives messages which are common across all contexts. For example, whenever the agent moves to the key location in *any* context, it receives the *same* message "You see here a key named The Master Key of Thievery". Similarly, when picking up the key, it receives the same message, "g - a key named The Master Key of Thievery." Here, we define the $\psi$ feature extractor in equation 7 to extract the message rather than the $(x, y)$ coordinates. Results in Figure 6b show that both the global and episodic bonuses work well for $|\mathcal{C}| = \infty$. This can be explained by the fact that regardless of the context, a policy which activates both of these messages aligns with the optimal policy. A global bonus will encourage the agent to explore diverse messages throughout training, eventually activating the two messages above which bring it close to the optimal policy. An episodic bonus will encourage the agent to activate diverse messages within each *episode*, which in this case similarly brings it close to the optimal policy.

### 3.3 COMBINING GLOBAL AND EPISODIC BONUSES

Taken together, the above examples suggest that global and episodic bonuses have complementary strengths and weaknesses, and that their effectiveness depends on the amount of shared structure among different contexts in the CMDP. At one end of the spectrum, singleton MDPs have complete sharing of structure among contexts (since they are all identical), and global episodic bonuses are best suited. At the other end of the spectrum, CMDPs such as the `MultiRoom` environment share little structure between contexts and episodic bonuses are preferred. The `KeyRoom` environment sits somewhere in between, where contexts are different but there is nevertheless shared structure between them, and both types of bonus are effective.

This raises the question: is there a bonus which works well across all of these different settings? We hypothesize that multiplying the episodic and global bonuses together would be more effective than

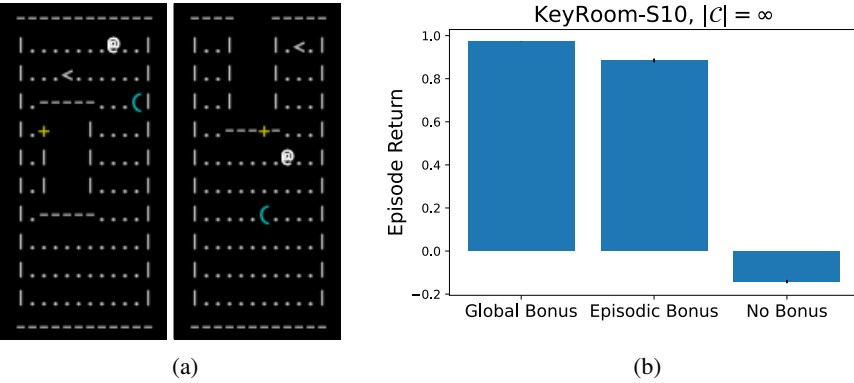

(a)                                                    (b)

Figure 6: a) Two example maps for `MiniHack-KeyRoom-S10` enviroment. `@` indicates agent, `(` key, `+` door, `<` start location and `>` goal. b) Performance of agents trained with the global bonus and the episodic bonus. Error bars represent the standard deviation across 5 random seeds.

either of them alone, and would work well across a wide range of CMDPs with differing amounts of common structure across episodes. The resulting combined bonus is given by:

$$b_{\text{combined}}(s_t) = \mathbb{I}[N_e(\psi(s_t)) = 1] \cdot \frac{1}{\sqrt{N(\psi(s_t))}} \tag{8}$$

This is motivated by the following observations. First, let us consider the MDP in Figure 4: note that following any of the corridors will maximize the episodic bonus by providing an episodic bonus of 1 at each step. The total combined bonus in equation 8 is then equal to the global bonus, and optimizing the global bonus causes the agent to visit each of the corridors until it reaches the one with the reward, solving the MDP.

Now let us consider the `MultiRoom` environment. If the agent is initialized roughly uniformly throughout the map, the global bonus will decay roughly uniformly across regions over time. This means that the bonus in equation 8 will be roughly equal to the episodic bonus (scaled by a constant), which we know is effective. Finally, as noted previously, in `KeyRoom-S10` both the episodic and global bonuses will assign high novelty to messages associated with picking up the key, which aligns with the optimal policy, suggesting that their product will also be effective.

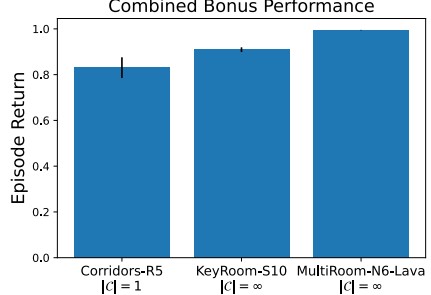

Figure 7: Performance of combined bonus. Error bars indicate standard deviation over 5 seeds.

Empirical results for all three environments are shown in Figure 7 (we use the same $\psi$ feature extractor as in previous experiments for each environment). We see that the combined bonus obtains good performance on all three environments, which suggests that it retains the advantages of both the global and episodic bonus.

## 4    DESIGN CHOICES FOR EPISODIC AND GLOBAL NOVELTY BONUSES

The previous section has shown that global and episodic bonuses succeed in different types of CMDPs, and that combining them via multiplication can yield a bonus which is more robustly effective. However, in order to facilitate interpretability we used count-based bonuses, which do not scale to complex, high-dimensional environments unless task-specific prior knowledge is used (e.g. knowing to extract $(x, y)$ positions or messages). In this section, we investigate whether our insights still hold when using more general bonuses which do not assume such prior knowledge. We

do this through a study of global and episodic bonuses based on function approximation across a wide range of tasks from the MiniHack suite (Samvelyan et al., 2021).

## 4.1 EXPERIMENTAL SETUP

As our experimental testbed, we use 18 procedurally-generated tasks from the MiniHack suite (Samvelyan et al., 2021) used in prior work (Henaff et al., 2022). The MiniHack tasks are designed to precisely evaluate different capabilities of a given agent, such as navigation, planning or the ability to use objects. Furthermore, many of the MiniHack tasks involve sparse rewards (which are only given on completion of the task) and complex observations which include irrelevant information. For evaluation, we follow the protocol suggested by Agarwal et al. (2021) and report the mean, median and interquartile mean (IQM) together with $95\%$ confidence intervals using stratified bootstrapping. We use 5 random seeds for each of the 18 tasks. Our full experimental details can be found in Appendix D.

## 4.2 RESULTS

We now investigate combining different global novelty bonuses from AGAC, RND and NovelD with the elliptical episodic bonus. We use E3B's elliptical bonus as our episodic bonus instead of a count-based one, since prior work has shown that count-based bonuses either fail in complex environments, or are highly dependent on task-specific feature extractors (Henaff et al., 2022). In contrast, the elliptical bonus has been shown to work well across a wide range of environments without requiring task-specific prior knowledge.

Two questions we aim to answer are: i) which global bonus (if any) gives the most improvements when combined with E3B's elliptical bonus, and ii) which strategy is best for combining the two bonuses. To answer these, we consider the following algorithms:

$$b_{\text{E3B}\times\text{AGAC}}(s_t) = \left[\phi(s_t)^\top C_{t-1}^{-1}\phi(s_t)\right] \cdot D_{\text{KL}}(\pi(\cdot|s_t)\|\pi_{\text{adv}}(\cdot|s_t))$$

$$b_{\text{E3B}\times\text{RND}}(s_t) = \left[\phi(s_t)^\top C_{t-1}^{-1}\phi(s_t)\right] \cdot \|f(s_t) - \bar{f}(s_t)\|_2^2$$

$$b_{\text{E3B}\times\text{NovelD}}(s_t) = \left[\phi(s_t)^\top C_{t-1}^{-1}\phi(s_t)\right] \cdot \left[\|f(s_{t+1}) - \bar{f}(s_{t+1})\|_2^2 - c\|f(s_t) - \bar{f}(s_t)\|_2^2\right]_+$$

$$b_{\text{E3B}+\text{AGAC}}(s_t) = \left[\phi(s_t)^\top C_{t-1}^{-1}\phi(s_t)\right] + \beta D_{\text{KL}}(\pi(\cdot|s_t)\|\pi_{\text{adv}}(\cdot|s_t))$$

$$b_{\text{E3B}+\text{RND}}(s_t) = \left[\phi(s_t)^\top C_{t-1}^{-1}\phi(s_t)\right] + \beta\|f(s_t) - \bar{f}(s_t)\|_2^2$$

$$b_{\text{E3B}+\text{NovelD}}(s_t) = \left[\phi(s_t)^\top C_{t-1}^{-1}\phi(s_t)\right] + \beta\left[\|f(s_{t+1}) - \bar{f}(s_{t+1})\|_2^2 - c\|f(s_t) - \bar{f}(s_t)\|_2^2\right]_+$$

Here $\phi$ is learned online using an inverse dynamics model. The algorithms above include all possible combinations of global bonuses (second term) with the elliptical bonus (first term), and combining the two by multiplication or by taking a weighted sum. For the algorithms which take a weighted sum, we tuned the $\beta$ term on a subset of tasks, and report the best value on all 18 tasks. We compare to E3B as a baseline since it was previously shown to outperform other methods such as IMPALA, RND, ICM, RIDE and NovelD (Henaff et al., 2022).

Results are shown in Figure 8. First, we see that additively combining any of the global bonuses with the elliptical episodic bonus does not provide a meaningful improvement over E3B for any metric. However, multiplicatively combining E3B with either RND or NovelD bonuses produces a large and statistically significant improvement in both median and IQM performance over E3B (the more robust metrics according to Agarwal et al. (2021)), as well as a modest improvement in mean performance. This establishes a new state-of-the-art on MiniHack.

One explanation for the superior performance of the multiplicative combination over the additive one is that the scale of the global bonus decreases significantly throughout training whereas the scale of the episodic bonus does not, since it is reset each episode. Because of this, if we combine the two bonuses via addition, the combined bonus will become increasingly dominated by the episodic

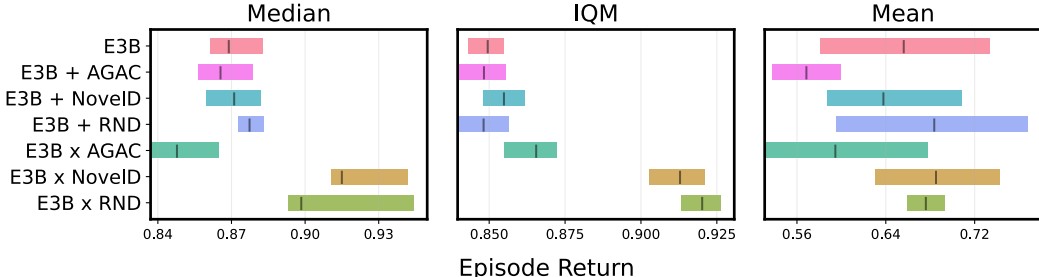

Figure 8: Aggregate performance on 18 MiniHack tasks. Bars indicate 95% confidence intervals computed using stratified bootstrapping.

bonus. However, if we are combining the two multiplicatively, the global bonus will still have an effect regardless of its scale. See Appendix E for additional results and discussion.

## 5 RELATED WORK

Exploration in singleton MDPs is a well-studied problem in RL, we give a more detailed overview of existing methods in Appendix C. However, these methods are designed for the singleton MDP setting and use some form of global bonus which, as we show in Section 3, is not always appropriate to the more general CMDP setting we consider here. More recently, RIDE (Raileanu & Rocktäschel, 2020), AGAC (Flet-Berliac et al., 2021) and NovelD (Zhang et al., 2021b) have begun to tackle exploration in procedurally-generated MDPs, a type of CMDP commonly used in empirical research. These methods use combinations of global bonuses designed for singleton MDPs and count-based episodic bonuses. The recent work of Henaff et al. (2022) highlighted the practical importance of these episodic bonuses, as well as the weaknesses of count-based episodic bonuses, and proposed the elliptical episodic bonus as a solution, but did not include a global bonus. Compared to these prior works, our work makes two contributions. First, whereas previous works justified using global bonuses by appealing to intuitions from singleton MDPs, and provided little justification for using episodic bonuses aside from their empirical performance, we provide deeper justifications for the use of each bonus in CMDPs. In particular, we examine in detail the behavior of global and episodic count-based bonuses across different representative settings, and shed light on how each bonus can drive exploration depending on the amount of shared structure across episodes. Second, whereas previous works have investigated different combinations of bonuses in isolation, there has not been a systematic comparison of bonuses and combination strategies, which we do in Section 4. This investigation results in a new algorithm which outperforms the previously proposed ones. Furthermore, our algorithm's multiplicative combination of global and episodic bonuses is well justified based on our previous investigations in Section 3.

## 6 CONCLUSION

In this work, we have taken steps towards better understanding the roles of global and episodic exploration bonuses in CMDPs. First, we study illustrative examples in which count-based versions of each bonus succeed and fail, showing that the two bonuses have complementary strengths and weaknesses. In particular, our experiments suggest that the effectiveness of each bonus depends on the amount of structure which is shared across episodes in the CMDP, and that episodic bonuses are more effective when there is little shared structure, while global bonuses benefit from more shared structure. We also show that combining global and episodic bonuses multiplicatively leads to increased robustness across different settings. Motivated by these observations, we perform an in-depth empirical study of approaches which combine global and episodic bonuses in the function approximation setting. This results in a new algorithm that sets a new state-of-the-art across 18 tasks from the MiniHack suite. Future research directions include more precisely understanding and characterizing the settings where each bonus will succeed, which may lead to improved algorithms which can adapt automatically to the setting at hand.

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
