# OpenReview forum: "Integrating Episodic and Global Novelty Bonuses for Efficient Exploration"
_ICLR.cc/2023/Conference — Submitted to ICLR 2023_

### Official Review · Reviewer_xEHz · 2022-10-25

**Confidence:** 3
**Correctness:** 3
**Technical Novelty And Significance:** 3
**Empirical Novelty And Significance:** 2
**Recommendation:** 6

**Clarity, Quality, Novelty And Reproducibility:**

Clarity: high. well written.
Quality: medium. meaningful questions, but the contributions are very marginal.
Novelty: not high
Reproducibility: medium. I couldn't find any information on tuning \beta parameters for 18 different tasks.

**Strength And Weaknesses:**

Strengths
- This work is based on the contextual MDP (CMDP) setting, where C is the context space in addition to the MDP setting. Per each episode, the context c is sampled from the context distribution.
- Nice and detailed explanations on background on exploration bonuses (RND, novelD, AGAC, RIDE) in section 2.2.
- I think the main part of this paper is the section 3, raising the questions of when are episodic bonuses and global novelty bonuses are useful, respectively.
- There are some example of CMDPs where global novelty bonuses fail, and advantage of the episodic bonuses succeed.(MiniHack, for example). Where the environment changes each episode, global bonuses fail (Figure 2).
- There are other cases of CMDPs where global novelty bonuses are preferred. This is useful (1) singleton MDP case (2) although with large |C|, the structures are shared across contexts.
- I think the arguments made by the authors are compelling so far.
- The experiments in the section 3 are valid
- Evaluations are valid.

Weakness
- Although the authors showed the insights on when to prefer global bonuses, and when to prefer episodic bonuses, and its relationship with level of structures being shared across the context. I think the contribution is not largely significant. This paper does not introduce a new algorithm,  although it shows some design choices for episodic and global novelty bonuses.
- Although the authors argue that general CMDPs with large |C| with shared structures across contexts are good fit for global bonuses, they didn’t add theoretical insights in it; only tried to prove it via one example.

**Summary Of The Paper:**

This work aims to analyze the use of two exploration novelty bonuses - global bonuses and episodic bonuses - in the problem of exploration in environments. Previous works show that global novelty is computed using agent’s entire training experience while episodic novelty is constrained within the specific episode the agent is in. This paper delves into the different behaviors of these two bonuses. When there is little shared structure between environments, episodic bonuses are useful; when there is more structure shared between environments, global bonuses are useful. This leads to the combination of global and episodic bonuses, and evaluation on various settings.

**Summary Of The Review:**

The idea is straightforward and convincing, but I think theoretical contributions / novelty / experimental evaluations are limited. I give the marginally below the threshold score, but I'm happy to raise the scores after the discussion with other reviewers.

---

> ### Author Response · Authors · 2022-11-17
> **Response**
>
> Thank you for the helpful review. We are glad you found our paper well written with meaningful questions, compelling arguments, and valid experiments, with straightforward and convincing ideas. It seems your main concerns are: 1) no theoretical insights, and 2) limited novelty.
>
>
> *Theoretical Insights*
>
> Regarding 1), this is not a theoretical paper and our aim isn’t to provide theoretical proofs, which we do not claim to do. Instead, we focus on gaining insights directly from empirical experiments on a number of challenging environments which are widely used in the literature. Given the large gap between theory and practice in CMDPs, we believe empirical insights could be more informative for researchers and practitioners at this point. Indeed, CMDPs such as MiniHack or Habitat are difficult to analyze because their structure cannot be described in simple terms, and a theoretical analysis of CMDPs would likely require strong simplifying assumptions which may not be realistic. We would like to note that our work is to our knowledge the first to provide insights into the relative strengths and weaknesses of global and episodic bonuses, and we believe these insights could motivate future theoretical work by the community in this area. It is often the case that phenomena are first investigated empirically, and only later understood from a theoretical perspective.
>
> *Novelty*
>
> Regarding 2), our insights and results are new, which is in itself a novel contribution. In fact, our paper is the first to study the strengths and weaknesses of global and episodic bonuses, and how we can combine them to get the best of both worlds. Our study shows that by combining episodic and global bonuses in a specific way, we can obtain significantly better results than prior methods or other combinations of the two types of bonuses. Even if each component of our algorithm was proposed in prior work, this particular combination results in a simple and highly effective algorithm (that can easily be applied to base RL algorithms), which we believe would be of interest for the community. Finally, our insights open the door to many future research directions such as: better understanding the global vs. episodic bonuses from a theoretical perspective, finding ways to automatically determine the optimal balance of global vs. episodic novelty bonuses, designing new measures of global and episodic novelty, to name a few.
>
>
> *Reproducibility*
>
> We added more information about how the \beta parameter was tuned in Appendix D.2. We will open source the code upon acceptance as well.
>
> *Additional experiments on pixel-based Habitat*
>
> In addition, we have now added new experiments on Habitat in Section 3.1 which shows that our conclusions for when episodic bonuses are useful are not specific to MiniHack, and also apply to settings with realistic maps and rich pixel-based observations.
>
> Thank you again for the valuable feedback. We hope that the clarifications above and the changes we made are sufficient for you to reconsider your assessment of the paper. If you have any outstanding concerns that prevent you from recommending acceptance, please don’t hesitate to let us know so we can discuss them in due time.

---

> > ### Author Response · Authors · 2022-12-05
> > **have we addressed your concerns?**
> >
> > Dear reviewer xEHz
> >
> > We hope you have had time to read our response, where we have added additional details about tuning the \beta parameter as requested, discussed the novel aspects of our work and added new experiments on Habitat. Also, you may find our discussion with review f9fK of interest, where we describe more precisely what we mean by shared structure across contexts. We hope that given these updates, you will consider raising your score. If not, please let us know which remaining concerns you have so we can discuss them. Thanks!

---

> > > ### Comment · Reviewer_xEHz · 2022-12-07
> > > **Rasing score from 5 to 6**
> > >
> > > After reviewing the responses from authors and having discussions with other reviewers and area chair, I decided to raise the score from 5 to 6.

---

### Official Review · Reviewer_f9fK · 2022-10-25

**Confidence:** 2
**Correctness:** 1
**Technical Novelty And Significance:** 2
**Empirical Novelty And Significance:** 3
**Recommendation:** 5

**Clarity, Quality, Novelty And Reproducibility:**

The paper is well written and the authors introduce a reasoning for each of their steps, that is clear to follow. The notation cloud use some slight improvements by clearly stating which methods (Sec. 2.2) are working on a featurized representation of the state ($\psi(s)$), but this is not required to fully understand the paper. The result section is missing some details, that the read can only try to infer. Namely, the state feature space used for the experiments in Sec. 3.3 and 4. Furthermore, Figure 6 lacks a frame of reference - meaning results without a combination of bonus terms. The reviewer assumes that the experiments shown in Figure 6 are based on the same feature space as Fig 4 & 5. Under that assumption, it seems like the global-only variant is outperforming the combined method in Corridors and KeyRoom, putting the claims into question. Something comparable is true for Section 4: Assuming that the message feature space was also used where applicable, some improvements are expected by combining E3B with a global method, because the global method should be capable of generalizing across contexts (compare Fig 5b). A comparison with a pure global method is required for substantiating the claims. It is still interesting that only the multiplicative method is able to show these improvements, but this cannot overcome the mentioned issues.

The analysis itself is novel, but the proposed method (multiplicative bonus) is a straight forward combination of existing ideas. However, additional insight into exploration strategies is a relevant topic in this domain, as this is important for reducing sample requirements and improving applicability. Reproducibility is assumed, given the source code should be available and architectural details as well as hyperparameters are given.

**Strength And Weaknesses:**

The authors try to tackle an interesting problem in a clear to understand way. The evaluation is reasonable diverse, but could be improvement by adding different domains (not just variants of MiniHack). However, the paper suffers from flaws in the reasoning, combined with a lack of a crucial detail (the used feature space, see next section). The claims from Sections 3.1 do only hold under the used state feature spaces, because it does not generalize due to the context dependent value of the feature space. It is likely, that a context independent feature space - like the message features from Section 3.2 - will not be affected by the mentioned issue. This is also supported by Figure 5b, showing competitive results for the episodic variant. Furthermore, the results from Sec. 3.3 and 4 are not clearly showing the assumed benefit, as explained in the next Section.

**Summary Of The Paper:**

The authors evaluate the relation between episodic and global novelty bonus with regard to contextual MDPs. They show two simplified examples for demonstrating the different advantages of the two novelty bonus types. Additionally, they explore 6 different combinations of the episodic E3B bonus with global novelty bonus definitions, proposed in the literature. The methods are evaluated on 18 MiniHack domains.

**Summary Of The Review:**

The paper is well written, but lacks a crucial detail and contains some questionable argumentations. Namely, the effect of the feature space is not sufficiently evaluated and it is not clear if a global-only bonus would be sufficient, given a context independent state feature space. In conclusion, the quality of the proposed method is not substantiated. The authors should clarify the mentioned feature space issues and add additional experiments with a global-only baseline and evaluate different feature spaces, if possible.

---

> ### Author Response · Authors · 2022-11-17
> **Response 2/2**
>
>
>
> - _Something comparable is true for Section 4: Assuming that the message feature space was also used where applicable, some improvements are expected…A comparison with a pure global method is required for substantiating the claims._
>
> Again, in Section 4 none of the methods use hardcoded features. We only use hardcoded features in Section 3, because they facilitate interpretability. Section 4 is designed to test whether the insights from Section 3 still hold when we do not use hardcoded features, and the improvements we observe suggest this is the case. Also, as noted in Section 4.2, E3B (which we use as a baseline) has already been reported to outperform pure global methods such as ICM and RND in [1].
>
> - _The evaluation is reasonable diverse, but could be improvement by adding different domains (not just variants of MiniHack)_
>
> We have added results on Habitat to Section 3.1. Habitat is a photorealistic simulator of indoor spaces used in embodied AI research. This is an environment which is conceptually similar to MultiRoom in the sense that at each episode, the agent finds itself in a different indoor space consisting of connected rooms, but the maps are much more complex and the observations are pixel-based. Here we observe similar results as for MiniHack-MultiRoom: the global bonus performs poorly, but the episodic bonus performs well.
>
>
> We hope we have addressed your concerns by clarifying the feature extractors we use and by adding additional results on Habitat. If you have any remaining questions or concerns that would prevent you from raising your score, please do not hesitate to let us know and we will do our best to address them.
>
> References:
> [1] Henaff, Raileanu, Jiang, and Rocktaschel. Exploration via elliptical episodic bonuses. NeurIPS, 2022.

---

> > ### Comment · Reviewer_f9fK · 2022-11-24
> > **Response to authors**
> >
> > Thanks for the clarification and additional results. I acknowledge, that i misunderstood the experiment setting in Sec. 4, however, i'm still not sure what the feature space is in that case, considering MiniHack allows for different observation types. However, this is only a clarity issue. The habitat results and [1] are also additional evidence relaxing my criticism concerning the global bonus.
> > Although, I'm still not satisfied concerning the feature spaces in Sec. 3. My choice of words was not optimal, therefore, i will try again to explain the reason for my doubts: Position features do not generalize across contexts, because the same position my have a different semantic. However, message features do generalize across contexts, because they are the semantic. Therefore, the advantages of the different bonus types can be reduced to the choice of feature and not necessarily to the domain.
> > Resultingly, i'm willing to raise my score but i still don't see a clear accept.

---

> > > ### Author Response · Authors · 2022-11-29
> > > **Response 2/2**
> > >
> > >
> > >
> > > We will now clarify what we mean by "structure" here, by defining it in terms of shared structure in the value functions across different contexts. We believe this will address your concern about the effect of feature encoding. Let us consider some state abstraction function $\psi: \mathcal{S} \rightarrow Z$ which could be hardcoded or learned. For example, it could extract $(x,y)$ positions (as described in Section 3.1) or messages (as described in Section 3.2), or it could be learned with an inverse dynamics model or other SSL method as done in Section 4. Now, define the value function $V_{\psi, c}: Z \rightarrow [R_{min}, R_{max}]$ as the expected discounted cumulative return under the optimal policy for each abstract state in $Z$ given $\psi$ and a context $c$.
> > >
> > > Now, let’s consider how much the value function $V_{\psi, c}$ changes across different contexts $c$. If it changes a lot, we will say the different contexts share little structure and if it changes little, we will say the different contexts share a lot of structure. Note that importantly, this value function depends _both_ on the MDP itself as well as the feature extractor $\psi$.
> > >
> > > To make this concrete, we can illustrate with some examples from the paper.
> > >
> > > **Example 1:** Consider the MultiRoom environment from section 3.1, where $\psi$ extracts $(x, y)$ positions. Here, $Z$ is a 2-d lattice of all possible $(x, y)$ positions, i.e. $Z = \psi(S) =$ {  $(x, y) \in \mathbb{N} \times \mathbb{N}: 1 \leq x \leq width, 1 \leq y \leq height$}. Now for each context $c$, $V_{\psi, c}$ will be high at the goal location and decrease outwards. Since each context has a different goal location, this means that $V_{\psi, c}$ will change a lot from one context to the next. This is what we mean by there being little structure shared across contexts.
> > >
> > > **Example 2:** Let us now consider the KeyRoom environment from Section 3.2, where $\psi$ extracts messages. Here, $Z$ is the set of all possible messages, i.e. $Z$ = {" ", "You see here a key named The Master Key of Thievery", "g - a key named The Master Key of Thievery"}. Here, the value function $V_{\psi, c}$ will _not_ change much across different contexts $c$, since for any context, visiting and picking up the key (which trigger messages 2 and 3 respectively) indicate high value regardless of the map configuration. This is what we mean by lots of structure being shared across contexts.
> > >
> > > Note that for the _same_ CMDP, if we use _different_ feature encodings, this can affect the amount of shared structure among value functions corresponding to different contexts. To illustrate, let's consider a third example:
> > >
> > > **Example 3:** Let us again consider the MultiRoom environment from Section 3.1, but now $\psi$ extracts messages instead of $(x,y)$ positions. Here, $Z$ is the set of all possible messages, i.e. $Z$ = {" ", "The door resists!", "The door opens"}. Messages 2 and 3 appear when the agent attempts to open a closed door connecting two rooms. To reach the goal, the agent needs to open 5 doors, hence these two messages indicate progress towards the goal and will have high value under $V_{\psi, c}$, regardless of the context $c$ which determines the map configuration. Unlike in example 1, here the value function $V_{\psi, c}$ will change little across contexts $c$, whereas when $\psi$ extracted positions, it changed a lot. This shows that our notion of shared structure depends on the feature encoding as well as the CMDP itself.
> > >
> > > For the above example, we also ran experiments with the global bonus and found that the global bonus worked well here. This further validates our theory that global bonuses perform well when there is lots of shared structure.
> > >
> > > We would like to note that in many cases, we have limited control over the features available to us. For example, although using message features enables the use of the global bonus for MultiRoom, there are other variants of MultiRoom in MiniHack where all the doors are open and messages are meaningless. More generally, features will often be learned and it is hard to guarantee they will induce value functions with shared structures across contexts. Therefore, it is still important to have bonuses which are effective with different amounts of shared structure, which the multiplicative bonus helps with.
> > >
> > > We will update the paper for the final version to incorporate these explanations. We hope we have addressed your remaining concerns, and that you will consider updating your recommendation to accept. If not, please let us know and we will further do our best to address them.

---

> > > > ### Author Response · Authors · 2022-12-05
> > > > **has this addressed your concerns?**
> > > >
> > > > Dear reviewer f9fK,
> > > >
> > > > We wanted to follow up and see if you had a chance to read our latest response, which we hope has addressed your remaining concerns about the feature spaces and their relationship to the effectiveness of episodic vs. global bonuses. If so, we would greatly appreciate if you consider raising your score to reflect this. If not, please let us know as soon as possible so we can further discuss. Thanks!

---

> > > ### Author Response · Authors · 2022-11-29
> > > **Response 1/2**
> > >
> > > Thanks for updating your score, we are glad some of your concerns have been resolved and we really appreciate you taking the time to read our response. We will do our best to address your remaining concerns.
> > >
> > > - _however, i'm still not sure what the feature space is in that case, considering MiniHack allows for different observation types_
> > >
> > > Indeed, MiniHack and NetHack allow for different observation types: symbolic image maps, statistics vectors and messages. We use all three modalities for both the policy networks and the $\phi$ encoders in Section 4, using the default architecture from the original MiniHack repo. Details can be found in Appendix C.1 but we will also clarify this for the final version of the paper.
> > >
> > > - _Although, I'm still not satisfied concerning the feature spaces in Sec. 3. My choice of words was not optimal, therefore, i will try again to explain the reason for my doubts: Position features do not generalize across contexts, because the same position my have a different semantic. However, message features do generalize across contexts, because they are the semantic. Therefore, the advantages of the different bonus types can be reduced to the choice of feature and not necessarily to the domain. Resultingly, i'm willing to raise my score but i still don't see a clear accept._
> > >
> > > Thank you for further clarifying your concerns, we believe we understand them better now. You are correct that the choice of features will have an influence on whether global or episodic bonuses are more useful. This is consistent with our discussion in the first paragraph of Section 3.3, about how episodic bonuses appear to work better in settings where there is little shared structure across contexts and global bonuses appear to work better when more structure is shared. An important thing to note, which may not have initially been clear and we will clarify in the final version of the paper, is that our notion of structure depends on the feature extractor as well as the CMDP itself.

---

> ### Author Response · Authors · 2022-11-17
> **Response 1/2**
>
> Thank you for the useful comments. We are glad you found our paper well written, the problem interesting, and the topic important for reducing sample requirements and improving applicability. It seems your main concerns relate to the effects of the features. We believe these are due to misunderstandings of our experiments and results, which we hope to address below.
>
>
> To answer your specific points:
>
> - _The claims from Sections 3.1 do only hold under the used state feature spaces, because it does not generalize due to the context dependent value of the feature space. It is likely, that a context independent feature space - like the message features from Section 3.2 - will not be affected by the mentioned issue._
>
>
> We believe there is a misunderstanding. In Section 3, both feature extractors which we use are context independent: the feature extractor is the same regardless of the context. For the navigation-based tasks (MultiRoom and Corridor), we extract the (x,y) position of the agent (regardless of the map/context). For the KeyRoom task, we extract the message (again, regardless of the map/context).
>
> The reason we use count-based bonuses with hardcoded features in Section 3 is to facilitate interpretability,  since reasoning precisely about bonuses learned with RND, NovelD, E3B etc is a lot harder. Using count-based bonuses on the raw state is not effective (since each observation is usually seen at most once), so for each task we chose features which align with the task reward. For navigation-based tasks (e.g. MultiRoom and Corridors), we used (x,y) positions, whereas for the KeyRoom task which is more skill-based, (i.e. it requires behaviors beyond simple navigation such as picking up objects, which are reflected by messages), we used message features. Please refer to [1] which contains a more detailed discussion of different feature extractors if this is not clear.
>
> We would also like to clarify that we do *not* use any hardcoded features for our experiments in Section 4. Whereas Section 3 aims to provide insights into the behavior of episodic and global bonuses (using count-based bonuses), Section 4 investigates the two types of bonuses empirically on a large set of tasks in the function approximation setting, using methods which do not rely on counts or features.
>
> Regarding the reviewer’s hypothesis that message features would work better:  in general, message features do not work on the MiniHack MultiRoom environments - Figure 2 of [1] shows that the message features fail on a similar MultiRoom environment when used with NovelD. We have confirmed this for the count-based bonus alone as well.
>
> - _The reviewer assumes that the experiments shown in Figure 6 are based on the same feature space as Fig 4 & 5. Under that assumption, it seems like the global-only variant is outperforming the combined method in Corridors and KeyRoom, putting the claims into question._
>
> We are not claiming that the combined bonus always outperforms the other two. Our first claim is that episodic and global bonuses work best on _different_ types of tasks, i.e. episodic bonuses work better on tasks with little shared structure, and global bonuses work better on tasks with lots of shared structure (including singleton MDPs). However, neither of the two works well across all types of tasks: global bonuses can fail on tasks with little shared structure (as shown in Figure 2), and episodic bonuses can fail on tasks with lots of shared structure (as shown in Figure 4b). Our second claim is that the combined bonus works more consistently than either the global bonus or episodic bonus alone across the different types of tasks. This is shown in Figure 6, where we see that the combined bonus obtains good performance (>80% success) across all three tasks considered. Even though the combined bonus might be slightly worse than the global or episodic bonus on *some* of the tasks, the important point is that it performs reasonably well across *all* of them.
>
> - _The result section is missing some details, that the read can only try to infer. Namely, the state feature space used for the experiments in Sec. 3.3 and 4._
>
> We have clarified that we are using the same feature extractors in Section 3.3 as in previous experiments. Furthermore, as mentioned above we do not use any hardcoded feature extractors in Section 4.
>
> References:
> [1] Henaff, Raileanu, Jiang, and Rocktaschel. Exploration via elliptical episodic bonuses. NeurIPS, 2022.

---

### Official Review · Reviewer_b6oF · 2022-10-26

**Confidence:** 3
**Clarity, Quality, Novelty And Reproducibility:** The paper is clearly written.
**Correctness:** 3
**Technical Novelty And Significance:** 2
**Empirical Novelty And Significance:** 2
**Recommendation:** 5

**Strength And Weaknesses:**

Strength:
1. The paper tackles the problem of episodic and global novelty, which is a very interesting setting.
2. The paper is well written.
3. The baselines are compared thoroughly.
4. The experiments is conducted over 18 tasks.

Weakness:
1. I feel the paper may dismiss some baselines like Never Give Up, MADE.
2. I feel like the experiments conducted are toy and it is hard for me to judge whether this is a valuable contribution in general. I would suggest conducting more experiments in different domains to support the conclusion.

**Summary Of The Paper:**

The paper tackles the problem of exploration in RL. This has long been a challenging problem in RL, the paper talks about the episodic novelty and global novelty in the exploration. The paper studies the two novelty and draws the conclusion that they are effective in different settings. Episodic novelty is most effective when there are structures shared while global novelty is most effective when there are fewer structures shared. The paper conducts experiments on 18 MiniHack tasks.

**Summary Of The Review:**

Overall I appreciate the authors' effort in tackling the exploration problem. However, it is a little hard for me to judge whether the conclusion can be drawn only from the MiniHack experiments. Please see the strengths and weaknesses above.

---

> ### Author Response · Authors · 2022-11-17
> **Response**
>
> Thank you for the helpful review. We are glad you found our paper well written, the baselines compared thoroughly, and that you agree that studying the problem of global and episodic novelty is very interesting. It seems your main concerns are: i) that we did not compare to Never Give Up (NGU) or MADE, and ii) we consider the MiniHack tasks, which may seem too simple.
>
> *Not compared with NGU or MADE*
>
> Concerning i) although we did not compare to NGU, others have reported that it performs poorly on MiniGrid (see https://openreview.net/pdf?id=j3GK3_xZydY), a related CMDP environment (several of the MiniHack tasks we consider are based on MiniGrid maps, but include additional challenges). The main difference between NGU and the E3BxRND variant we compare to in Section 4 is that NGU uses a different episodic bonus. Specifically, NGU’s episodic bonus is based on the euclidean distance between the current observation’s embedding and its k nearest neighbors. A conceptual difference with E3B’s elliptical bonus is that the elliptical bonus adjusts the scale of each dimension whereas NGU’s euclidean distance does not. Therefore, if some features are on a much larger scale than others, they may dominate the bonus. We are currently prioritizing pixel-based experiments since they have been requested by several reviewers, but will add comparisons to NGU for the final version. Note that MADE requires estimating densities of state-action pairs, which is not obvious in high dimensions, whereas our E3B-based variant does not.
>
>
>
> *Only Evaluated on MiniHack*
>
> To address ii), we would first like to note that although the MiniHack tasks may seem simple from a _perception_ standpoint (since a lot of information is in symbolic form), they are in fact quite challenging from many other standpoints. In particular, the 18 tasks we use include many challenging features such as partial observability, stochastic dynamics, moving entities/distractors, as well as sparse rewards and generalization. In fact, strong methods such as RND/ICM/RIDE which succeed on pixel-based tasks such as Atari, Vizdoom, or Mario, fail on most of the MiniHack tasks we consider here, as shown in [1]. Therefore, we believe that our results over a wide range of MiniHack tasks already constitute strong empirical evidence for the conclusions we draw and the effectiveness of our approach .
>
> Nevertheless, we do agree with the suggestion that including results on pixel-based environments would strengthen the paper. We have included a new section with results for reward-free exploration on Habitat (using the same experimental setup as [1]), which involves rich pixel-based observations and maps with complex geometries. Habitat is conceptually similar to the MultiRoom environment shown in Figure 1, in the sense that at each episode, the agent finds itself in a new indoor space consisting of connected rooms, and each episode corresponds to a different map. However, the maps in Habitat are more complex than the ones in the MiniHack MultiRoom environment and the observations are pixel-based. Here we see trends that are consistent with those reported in Section 3.1 for MultiRoom: the episodic bonus (E3B) succeeds, whereas the global bonuses (RND, ICM and NovelD) perform much worse.
>
> We hope our response and paper updates have adequately addressed your concerns and that you will consider increasing your support for our paper. If you have any outstanding concerns, please don’t hesitate to let us know.
>
> References:
> [1] Henaff, Raileanu, Jiang, and Rocktaschel. Exploration via elliptical episodic bonuses. NeurIPS, 2022.

---

> > ### Author Response · Authors · 2022-12-05
> > **Added NGU baseline**
> >
> > We have run the requested NGU baseline and will add it to the final version of the paper. We found that it does not perform very well, consistent with  https://openreview.net/pdf?id=j3GK3_xZydY. The results aggregated over all 18 MiniHack tasks are below (we include E3B and IMPALA for reference):
> >
> > | Method | Median | IQM | Mean |
> > |------      | --------  | ---   | --- |
> > | E3B       | 0.856 | 0.843 | 0.645 |
> > | NGU     |  0.373 | 0.455 | 0.395 |
> > | IMPALA| 0.000 | 0.000 | -0.002|
> >
> > NGU improves over vanilla IMPALA (which has no intrinsic reward), demonstrating that it helps exploration, but it is still far from E3B, which is the baseline we compare the different methods against in Section 4.2.
> >
> > With the additional Habitat results and the NGU baseline, we believe we have addressed the two main concerns you raise in the weaknesses section, and so would greatly appreciate if you would consider raising your score. If there is any reason for not doing so, please let us know as soon as possible so we can discuss and try to address any remaining concerns.

---

### Official Review · Reviewer_2WV4 · 2022-10-31

**Confidence:** 3
**Correctness:** 3
**Technical Novelty And Significance:** 3
**Empirical Novelty And Significance:** 3
**Recommendation:** 6

**Clarity, Quality, Novelty And Reproducibility:**

The paper was clear and well written, on in a setting (contextual MDPs) that is important yet understudied, and it provides novel results concerning the use of exploration bonuses. I have not seen other investigations on combining global & episodic explorations besides this one, which is cool. However,  their investigations on combining exploration bonuses in this paper is a little scant even though it is one of their main novelties. This can be improved.

**Strength And Weaknesses:**

Strengths: this study provided nice toy tasks that gave good intuition about when global vs episodic exploration bonuses would be beneficial and when they would not. Their multiplicative combination solution at the end of the paper (combining the 2) was a simple solution that can be easily implemented. They hypothesized why it works (the relative scaling of gobal vs episodic bonuses over  episodes). Finally, the fact that they tried it both for count-based bonuses as well as the more scalable E3B bonus was commendable.

Weaknesses: The tasks tried remain toyish and are limited in scope (confined to the minihack suite). While I really liked the demonstrations they provided, and think that minihack is a good start, it would have been nice to see at least a hint of either 1) how global vs episodic exploration bonuses behave in a slightly more scaled up task (e.g. procgen or anything with nontrivial pixels), or 2) how their multiplicative solution behaves in such a scaled up task and whether it still works.

Finally, their investigation of their multiplicative solution ends on a hypothesis: that it works because of the relative scaling of global vs episodic bonuses over  episodes. It would have been great if they could provide some more comprehensive empirical evidence for this hypothesis. The question of how to combine global & episodic exploration bonuses is a good question (and more generally, understanding  the effects of exploration durign learning/generalization is a very important question in RL). The authors did a cool experiment investigating multiplicative vs additive combination.  It is a very simple manipulation, but simple manipulations often have great merit. They should have dissected this experiment  for a more comprehensive understanding of this result.

**Summary Of The Paper:**

This paper investigated the benefits of global vs episodic exploration bonuses, and provided interesting intuition as to when each would help learning and when it would not. Moreover, this paper studies setting of contextual MDPs, which is relatively understudied compared to single MDPs. It proposes a simple way of combining global and episodic bonuses that work nicely in the tasks that they investigate.

**Summary Of The Review:**

This paper was an interesting and easy read and contains useful and novel results and intuitions for the community concerning the use of exploration bonuses, especially their combination. However, as it stands, there is not yet a hint of how their study would behave in slightly more scaled up tasks, and their analysis on combining exploration bonuses is a little too scant. If these factors can be rectified, I would definitely move my score up on what I think is an interesting paper.

---

> ### Author Response · Authors · 2022-11-17
> **Response**
>
> Thank you for the insightful review, and we are glad you found that our paper contributes useful and novel results/intuitions for an important yet understudied problem. It appears the two main concerns are i) our use of the MiniHack tasks, which are not pixel-based and may appear too simple, and ii) the fact that our explanation for the superior performance of the multiplicative bonus remains a hypothesis.
>
> To address ii), we have included plots showing the evolution of the global and episodic bonuses throughout training (see Appendix E). We see that the global bonus does indeed decay significantly to very small values, while the episodic bonus stays within a relatively stable range that is higher than that of the global bonus.
>
>
> To address i), we would first like to note that although the MiniHack tasks may seem simple from a _perception_ standpoint (since a lot of information is in symbolic form), they are in fact quite challenging from many other standpoints. In particular, the 18 tasks we use include many challenging features such as partial observability, stochastic dynamics, moving entities/distractors, as well as sparse rewards and generalization. In fact, strong methods such as RND/ICM/RIDE which succeed on pixel-based tasks such as Atari, Vizdoom, or Mario, fail on most of the MiniHack tasks we consider here, as shown in [1]. Therefore, we believe that our results over a wide range of MiniHack tasks already constitute strong empirical evidence for the conclusions we draw and the effectiveness of our approach .
>
> Nevertheless, we do agree with the suggestion that including results on pixel-based environments would strengthen the paper. In Section 3.1 we have included new results for reward-free exploration on Habitat (using the same experimental setup as [1]), which involves rich pixel-based observations and maps with complex geometries. Habitat is conceptually similar to the MultiRoom environment shown in Figure 1, in the sense that at each episode, the agent finds itself in a new indoor space consisting of connected rooms, and each episode corresponds to a different map. However, the maps in Habitat are more complex than the ones in the MiniHack MultiRoom environment and the observations are pixel-based. Here we see trends that are consistent with those reported in Section 3.1 for MultiRoom: the episodic bonus (E3B) succeeds, whereas the global bonuses (RND, ICM and NovelD) perform much worse.
>
>
> We would also like to note that we previously tried running different exploration algorithms on ProcGen, but found that these did not offer clear improvements over vanilla policy gradient. This is likely due to the rewards in ProcGen being dense (although there is an exploration option for some of the tasks, this restricts the environment to a single seed for which the reward is sparse, so this reduces to a singleton MDP rather than a CMDP). We therefore do not think this is a useful benchmark for studying exploration in CMDPs.
>
> We hope that we have addressed both of your concerns, and that you will consider raising your score. Please let us know if there are any remaining concerns we can address.
>
> References:
> [1] Henaff, Raileanu, Jiang, and Rocktaschel. Exploration via elliptical episodic bonuses. NeurIPS, 2022.

---

> > ### Comment · Reviewer_2WV4 · 2022-11-24
> > **Response**
> >
> > Thanks to the authors for addressing these concerns. For me, the habitat experiments are a nice addition. Both of my concerns were resolved so I have raised my score by one grade.

---

### Author Response · Authors · 2022-11-18
**Summary of updates**

We would like to thank all the reviewers for their careful comments and feedback, and we have updated the paper as a result to address the comments. Specifically:

Reviewers **2WV4**, **b6oF** and **f9fK** all expressed that they would like to see additional tasks beyond MiniHack. We have added new experiments using the photorealistic pixel-based Habitat simulator to Section 3.1. This is a conceptually similar task to the MiniHack-MultiROom navigation task in the sense that at each episode, the agent finds itself in a different map consisting of connected rooms. However, the maps in Habitat are significantly more complex and the observations are pixel-based. Here we see similar conclusions as for the MiniHack-MultiRoom environment: the episodic bonus performs well and the global bonuses perform poorly. This provides evidence that our insights are not specific to MiniHack.

We have also added additional plots and analysis of the behavior of the global and episodic bonus in Appendix E, as requested by reviewer **2WV4**, which support our hypothesis why combining the bonus multiplicatively works better than combining them additively.

We have clarified our use of feature extractors in response to the comments from reviewer **f9fK**. In particular, the feature extractors used in Section 3 to facilitate interpretability are *not* context-dependent, and we do not use *any* hardcoded features for the methods in Section 4.

We believe we have addressed all the main concerns brought up by the reviewers. If there are any remaining concerns that stand between us and a recommendation for acceptance, please let us know and we will do our best to address them.

---

### Decision · Program_Chairs · 2023-01-20

**Decision:**

Reject

**Justification For Why Not Higher Score:**

Please see my comments in the "Summary of AC-reviewer meeting".

**Justification For Why Not Lower Score:**

N/A

**Metareview: Summary, Strengths And Weaknesses:**

This paper examines the use of global versus episodic novelty bonuses for exploration in multi-task RL, and identifies that these two types of bonuses are useful in different contexts. Controlled experiments are performed in a few carefully selected MiniHack levels to demonstrate this effect, as well as in a more realistic domain (Habitat). The paper then proposes a multiplicative combination of these bonuses, and shows that this performs comparably well across multiple different types of environments in MiniHack, outperforming each individual bonus type on average.

The reviewers agreed that the paper studies an important and interesting problem, that the experimental evaluation on MiniHack is extensive, and that the results are insightful. In the initial reviews, the reviewers expressed some reservations that the evaluation was limited to the MiniHack domain, and that some potential baselines were missing. The authors' rebuttal partially addressed these concerns; in particular, during the discussion the reviewers said that they appreciated the experiments with Habitat and NGU and found that they strengthened the paper. However, there was also a lack of confidence that the results of the paper would necessarily generalize beyond the MiniHack domain. While the Habitat experiment helps, it would be better if the paper also had results across more domains in Section 4 demonstrating the same effect (that the different bonuses individually do well on different types of tasks, and that their combination is robust) in different settings.

While I agree the paper is insightful, the big question I find myself asking is: does this insight---that global and episodic bonuses should be combined to achieve the best exploration performance in CMDPs---hold in other CMDPs besides MiniHack? The Habitat experiment provides support for part of the hypothesis regarding episodic bonuses, but there is otherwise no evidence to suggest that the insight will hold more generally. While this might be less of a concern with a more canonical environment, MiniHack has a somewhat unusual observation space (most RL environments don't contain text observations, for example), which makes me less sure about the robustness of the results. As such, I think the paper falls (slightly) short of the bar for publication at ICLR. It is a very promising hypothesis, though, and I encourage the authors to continue working on this and to submit a revision to a future venue.

**Summary Of Ac-Reviewer Meeting:**

I held a meeting with 3/4 reviewers (unfortunately there was not a time that all reviewers could attend). Two of the reviewers initially leaned slightly towards rejection, and one (plus myself) initially leaned towards acceptance. Points raised against the paper included: insufficient evaluation (i.e., on domains besides MiniHack), lack of novelty in the combined bonus, and lack of theoretical insights. Points raised in favor of the paper included the insightfulness of the paper, its scientific approach, and the addition of the Habitat experiment. The discussion did make some of the reviewers more positive about the paper, but also helped to clarify that the generality of the results was not fully convincing. Moreover, the reviewers felt that an empirical "scientific" paper such as this should meet a higher bar for empirical rigor compared to one that introduces a new method.

On the other hand, the MiniHack evaluation itself is already quite extensive (18 tasks) and they achieve SoTA results on this domain. However, on the other hand, I do agree with the point that the evaluation could be more extensive (ie on domains beyond MiniHack). MiniHack really only captures one part of the space of MDPs more generally, so even if the phenomena explored by the paper holds there, it's unclear how well it would generalize more broadly (e.g. to a continuous control task suite).